# Criteria and Non-Criteria Antiphospholipid Antibodies in Antiphospholipid Syndrome: How Strong Are They Correlated?

**DOI:** 10.3390/biomedicines11082192

**Published:** 2023-08-03

**Authors:** Simona Caraiola, Laura Voicu, Ciprian Jurcut, Alina Dima, Cristian Baicus, Anda Baicus, Claudia Oana Cobilinschi, Razvan Adrian Ionescu

**Affiliations:** 1Fifth Department-Internal Medicine (Cardiology, Gastroenterology, Hepatology, Rheumatology, Geriatrics), Family Medicine, Occupational Medicine, Faculty of Medicine, “Carol Davila” University of Medicine and Pharmacy, 050474 Bucharest, Romania; 2Internal Medicine Department, Colentina Clinical Hospital, 020125 Bucharest, Romania; 3Internal Medicine Department, “Dr. Carol Davila” Central University Emergency Military Hospital, 010825 Bucharest, Romania; 4Rheumatology Department, Colentina Clinical Hospital, 020125 Bucharest, Romania; 5Laboratory Department, The University Emergency Hospital, 050098 Bucharest, Romania; 6Rheumatology Department, “Sf. Maria” Clinical Hospital, 011172 Bucharest, Romania

**Keywords:** antiphospholipid syndrome, criteria, non-criteria, antiphospholipid antibodies, correlation

## Abstract

The place of non-criteria antiphospholipid antibodies (aPLs) in the diagnosis of antiphospholipid syndrome (APS) is still debatable. The aim of this research was to evaluate the correlations between the titres of non-criteria aPLs (anti-phosphatidylethanolamine (aPE), anti-phosphatidylserine (aPS), and anti-prothrombin (aPT) antibodies), and the ones of the already studied criteria aPLs (anti-cardiolipin (aCL) and anti-β2 glycoprotein I-aβ2GPI antibodies). Altogether, 72 APS (30 primary and 42 secondary) patients were included in our study. High correlation coefficients (rs) were found between aPS IgM and aCL IgM, overall (0.77, *p* < 0.01), as well as in the primary (0.81, *p* < 0.01), and secondary (0.75, *p* < 0.01) APS subgroups. Low or statistically insignificant correlations were observed between IgG/IgM isotypes of aPT and aCL, or aβ2GPI, in the entire study population, and when evaluating the subgroups. Therefore, moderate correlations were mainly identified between the tested non-criteria antibodies and the criteria ones, suggesting little added value for the use of the tested non-criteria aPLs, with the exception of aPT, which seems to have different kinetics and might be a promising APS diagnostic tool.

## 1. Introduction

Antiphospholipid syndrome (APS) is an autoimmune disorder defined by the occurrence of specific clinical events—namely venous, arterial, and small vessel thrombosis, or pregnancy pathology, such as recurrent abortions, foetus death, premature birth, and placental insufficiency—in patients with persistently positive criteria antiphospholipid antibodies (aPLs) [1,2,3].

Phospholipids play a significant structural part in cellular membranes, where they are organised under a characteristic bilayer structure [4]. Phosphatidylcholine (PC) and sphingomyelin (SM) are mainly found in the exoplasmic leaflet, whilst phosphatidylethanolamine (PE), phosphatidylserine (PS), and phosphatidylinositol (PI) are predominantly located in the cytosolic leaflet, generating an asymmetric configuration [5]. A particular phospholipid—cardiolipin—is located in the mitochondrial membrane, where it exerts essential roles in respiration, transmembrane transport, inflammation, or apoptosis [6]. Along with phospholipids, phospholipid-binding proteins and different coagulation factors represent targets of the autoimmune reaction in APS. A multitude of autoantibodies against these structures have been described in recent decades [3,7]. However, out of this extensive group, only lupus anticoagulant (LAC), IgG or IgM isotypes of anti-cardiolipin (aCL) antibodies, and IgG or IgM isotypes of anti-β2 glycoprotein I antibodies (aβ2GPI) are currently included in the APS classification criteria [1]. The discovery of new aPLs raised questions about their relationship with the APS clinical manifestations as well as about their possible role in the APS diagnosis. The detection of “non-classical” or “non-criteria” aPLs might be especially useful in the diagnosis of the so-called “seronegative APS” [8]—cases presenting with thrombotic events or obstetric morbidity but testing negative for criteria aPLs [9].

The aim of this research was to evaluate the correlations between six non-criteria aPLs, namely the IgM and IgG isotypes of anti-phosphatidylethanolamine (aPE), anti-phosphatidylserine (aPS), and anti-prothrombin (aPT) antibodies, with the IgM and IgG isotypes of criteria aCL and aβ2GPI, in order to identify aPLs with different kinetics than criteria antibodies, which might be used as new serological tools for APS diagnosis.

## 2. Materials and Methods

An observational, cross-sectional, single-centre study was conducted. Patients diagnosed with APS according to the 2006 Sydney classification criteria [1] and admitted to the Internal Medicine Department of Colentina Clinical Hospital were prospectively enrolled. The following exclusion criteria were applied: age under 18 years, concomitant acute infectious disease, sarcoidosis, pregnancy, or being in the first 6 months postpartum. According to the APS clinical presentation, two subgroups were defined from the original study group: patients diagnosed with primary APS, in whom APS diagnosis was not associated with another disorder; and patients in whom APS diagnosis was associated with other pathologies, such as autoimmune diseases (systemic lupus erythematosus (SLE), rheumatoid arthritis, Sjögren syndrome, systemic sclerosis, polyarteritis nodosa, mixed connective tissue disease, and polymyositis) and chronic hepatitis or various types of neoplasms, who were categorised in the secondary APS subgroup.

Demographic data were collected as well as data regarding the APS onset and clinical manifestations. Laboratory data with respect to APS serology at the time of the diagnosis and enrolment were analysed.

In all cases, blood samples were collected at inclusion, centrifuged, and stored at −70 °C. An aPL panel, comprising IgM and IgG isotypes of aCL, aβ2GPI, aPE, aPS, and aPT, was performed for each sample. Technical reasons did not allow testing the LAC, since the samples could not be processed as fast as the determination protocol required [10,11].

IgG and IgM titres of aCL, aβ2GPI, aPE, aPS, and aPT were detected using ELISA kits (Aesku Diagnostics, Wendelsheim, Germany), and the Chemwell 2910 Analyser (Awareness Technology, Palm City, FL, USA). For the utilised kits, the laboratory cut-off values are as follows: normal values are situated below 12 U/mL, the positive values are above 18 U/mL, and the values between 12 and 18 U/mL are considered equivocal.

Categorical variables were compared using the chi-square test. Odds ratio (OR) and 95% confidence intervals (CIs) were calculated to determine the likelihood of clinical events’ appearance as well as of aPL positivity in primary and secondary APS. Continuous variables having a parametric distribution are expressed as mean ± standard deviation, and the means were compared with Student’s *t*-test. For variables with a non-parametric distribution, the results are presented as median values (minimum–maximum), which were obtained using Mann–Whitney and Wilcoxon tests. The relation between the expression of different criteria and non-criteria aPLs was analysed using the non-parametric Spearman rank-order correlation. Spearman’s correlation coefficient (r_s_) measures the direction (+/−) as well as the strength of the monotonic association of two variables with non-parametric distribution [12]. Four degrees of correlation were defined: important (0.80–1.00), moderate (0.50–0.79), weak (0.20–0.49), and neglectable (0.00–0.19). ROC (receiver operating characteristic) curve analysis was performed in order to evaluate the utility of criteria and non-criteria aPLs in predicting the APS subtype and the risk of various specific APS clinical events. In order to reject the null hypothesis, a two-sided *p*-value less than 0.05 was interpreted as being statistically significant. SPSS 20.0 software (IBM Corp., Armonk, NY, USA) was used for the statistical analysis of the collected data.

Informed consent was obtained from all participants, and anonymity was entirely ensured for the collected data. The study was approved by the Clinical Research Ethics Committee of “Carol Davila” University of Medicine and Pharmacy Bucharest.

## 3. Results

Out of the 72 included subjects, 62 (86.1%) were females. The patients had a mean age of 45 years at the time of enrolment, while the mean age at diagnosis was 39 years. The duration of the disease had a median value of 3 years (Table 1).

From the entire study population, 30 patients were registered in the primary APS subgroup, whilst the remaining 42 were listed in the secondary APS subgroup. No statistically significant difference based on gender, age at enrolment, or age at diagnosis could be detected between the study subgroups. Nevertheless, the duration of the disease was significantly longer among patients diagnosed with secondary APS—8.55 versus 2.87 years, *p* < 0.001 (Table 1).

In the secondary APS subgroup, 28 patients had SLE. In 20 of these patients, the SLEDAI scores were mild (0–5). Six patients registered moderate scores (0–6), while only two patients had severe disease activity (SLEDAI scores 13–20). The median value of SLEDAI scores was 2 (0–22).

The most frequently registered APS clinical manifestation was deep venous thrombosis. Arterial thrombosis, obstetric morbidity, and the association of thrombotic events with pregnancy pathology were all more often encountered in the secondary APS subgroup. However, the differences were not statistically significant. The enrolled patients also had non-criteria clinical features, such as livedo, vegetation, and migraine, but no statistically significant differences were registered between the study subgroups regarding these manifestations. At diagnosis, LAC was positive in all the included patients. Double- and triple-positive aPL profiles were more frequently encountered in the secondary APS subgroup, although no significant differences were registered when compared with the primary APS subgroup. The main clinical and serological characteristics at the time of diagnosis of the study subgroups are presented in Table 2.

At the moment of enrolment, most of the aPLs proved to be below the laboratory cut-off for positive values (Table 3). Out of the tested panel, aPE and aβ2GPI were encountered on a more frequent basis. The positive results were more frequently registered in secondary APS patients. Moreover, triple- and quadruple-positive aPL profiles were almost exclusively found in the secondary APS subgroup (Table 3). Only one patient with a secondary APS diagnosis registered in the quintuple-positive aPL profile.

However, according to the laboratory cut-off, the tested aPLs were mainly negative, and the obtained titres had extremely varied values, as observed in Table 4. No significant differences in the titres were determined between the primary and secondary APS subgroups, except for aCL IgG, which registered significantly higher values in secondary APS patients (*p* = 0.03).

ROC analysis was performed in order to evaluate the utility of criteria and non-criteria aPLs in predicting the APS primary or secondary subtype. The largest area under the curve (AUC) was registered for aPT IgG (Appendix A) as 0.517 (0.377–0.658), suggesting aPT as the best predictor for the APS subtype.

The various specific APS clinical events found in the medical history of the patients were evaluated using the ROC analysis. When considering deep venous thrombosis, the most important predictor seemed to be aPS IgG, with an AUC of 0.463 (0.326–0.600). For the association of deep venous thrombosis with arterial thrombosis, the highest AUC value was found for aβ2GPI IgM, 0.445 (0.209–0.682), while aPT IgM seemed to be the best predictor for deep venous thrombosis associated with obstetric morbidity, as its AUC was 0.667 (0.401–0.933). The detailed results are presented in Appendix A.

As shown in Appendix A, when evaluating the value of aPLs in predicting arterial thrombosis, the highest AUC was registered for aβ2GPI IgM as 0.525 (0.382–0.668), closely followed by aPT IgG, with 0.504 (0.369–0.639). The best predictor for arterial thrombosis associated with obstetric morbidity proved to be aβ2GPI IgM, with its AUC having a value of 0.706 (0.572–0.841). For the association of arterial thrombosis with deep venous thrombosis and obstetric morbidity, high AUCs were detected for aβ2GPI IgG, with 0.728 (0.475–0.980), aβ2GPI IgM, with 0.706 (0.497–0.915), and aPS IgM, with 0.651 (0.172–1.000).

The best predictor of obstetric pathological events among the analysed antibodies proved to be aβ2GPI IgM (Appendix A), with an AUC of 0.650 (0.501–0.800), followed by aPT IgG, which registered an AUC of 0.631 (0.448–0.778).

As presented in Appendix A, the development of non-criteria manifestations of APS seems to be best predicted by aPE IgG, with an AUC of 0.569 (0.431–0.707).

Most of the obtained correlations between the overall analysed criteria and non-criteria aPLs proved to be statistically significant, with the majority of *p*-values set below 0.01, and the correlation coefficient was predominantly situated in the moderate interval (Table 5). The highest r_s_ was registered between aPS IgM and aCL IgM (r_s_ 0.77, *p* < 0.01), closely followed by the correlation of aPS IgG with aCL IgG (r_s_ 0.72, *p* < 0.01). The lowest correlation coefficients were found between both IgM and IgG isotypes of aPT and aCL IgG (r_s_ 0.25, *p* < 0.05; r_s_ 0.26, *p* < 0.05, respectively), and in the aPT IgG–aβ2GPI IgG pair (r_s_ 0.26, *p* < 0.05). No significant correlation was found between aPT IgM and aPS IgG (r_s_ 0.21, *p* = ns).

Statistically significant correlations were found in the primary APS subgroup between the criteria and non-criteria aPLs, with the r_s_ generally having moderate values. As observed in Table 6, aCL IgM correlated with many of the non-criteria aPLs to different degrees. The most important correlation was again observed between aPS IgM and aCL IgM (r_s_ 0.81, *p* < 0.01), followed by the correlation of aPS IgG with aCL IgG (r_s_ 0.76, *p* < 0.01). It is also important to mention that IgG and IgM isotypes of aPE registered moderate statistically significant correlations with most of the tested aPLs. No significant correlations were found for the IgM and IgG isotypes of aPT with the criteria aCL IgG (r_s_ 0.14 and r_s_ 0.25, respectively, *p* = ns) or aβ2GPI IgG (r_s_ 0.31 and r_s_ 0.27, respectively, *p* = ns). Moreover, aPT IgM insignificantly correlated once more with non-criteria aPS IgG (r_s_ 0.26, *p* = ns).

Significant correlations between criteria and non-criteria aPLs were observed in the secondary APS subgroup as well. The obtained coefficients are listed in Table 7. High r_s_ were found in the aPS IgM–aCL IgM (r_s_ 0.75, *p* < 0.01), and aPE IgM–aCL IgM (r_s_ 0.74, *p* < 0.01) pairs. Notably, aPT insignificantly correlated with criteria aPLs (aPT IgG and aβ2GPI IgM (rs 0.25, *p* = ns); aPT IgM and aβ2GPI IgG (rs 0.29, *p* = ns); and aPT IgM and aCL IgG (rs 0.27, *p* = ns)) as well as with non-criteria aPLs (aPT IgM/IgG and aPS IgG (rs 0.14 and rs 0.26, respectively, *p* = ns) as well as aPT IgM and aPE IgG (rs 0.18, *p* = ns)). Furthermore, no statistically significant correlations were found between aPS IgG and aβ2GPI IgG (rs 0.29, *p* = ns) or between aPE IgG and aCL IgG (rs 0.29, *p* = ns).

## 4. Discussion

Although the first notions about aPLs date back to the beginning of the 20th century, it was not until the 1980s—with the discovery of specific identification serological tests—that a thorough characterisation of these molecules was initiated [13]. Around the same time, data regarding the pathological specific events linked to the presence of aPLs, namely thromboses and pregnancy pathology, started to emerge [9,13,14]. In the 1990s, the intermediary role of the β2 glycoprotein I between aPLs and phospholipids was described [13]. Subsequent studies suggested that cases of aβ2GPI positivity in the absence of serum aCL were not frequently encountered [15]. In the early 2000s, seronegative APS captured the interest of the scientific community [8,15]. Regardless of the labelling, more recent research indicates that about 30% of the so-called “seronegative” APS cases might be positive for one or more non-classical aPLs. However, non-criteria aPLs are more frequently encountered in seropositive APS than in seronegative APS patients [16].

The importance of testing for non-criteria aPLs has been recently indicated by an impressive number of studies. The data obtained by Liu et al. and Hu et al. show that testing for non-criteria aPLs could facilitate APS diagnosis [3,17]. In addition, the detection of non-criteria aPLs might be helpful in the risk stratification of the patients diagnosed with APS and might contribute to preventing specific APS clinical events [18,19,20].

Despite the consistent efforts made in recent years in the attempt to update the APS classification criteria, and the increasing evidence supporting the relevance of non-criteria aPLs in the diagnosis and management of APS, this issue still requires further research. The lack of standardised testing and the difficult access to these investigations reinforce the arguments against the inclusion of non-diagnostic aPLs in the classification criteria [21].

The impossibility of detecting criteria aPLs does not necessarily mean a milder APS course. On the contrary, it has been suggested that the presence of non-criteria aPLs in the serum correlates with the severity of the disease [22]. Moreover, some APS events, such as miscarriage, were more frequent in patients with seronegative APS, when compared to seropositive APS patients [16]. According to Volkov et al., aPT IgG positivity is associated with arterial thrombosis, and the presence of aCL IgG and aPS IgG or the isolated detection of anti-annexin-5 (aAN) IgG seems to be correlated with APS-specific pregnancy pathology, while aPT IgG, anti-phosphatidylglycerol (aPG) IgG, anti-phosphatidylinositol (aPI) IgG, and aAN IgG were connected to CNS manifestations of the APS [23]. Similar findings were emphasised by Zhang et al., whose research found a significant association of aPS IgG and IgM, aPI IgG, and anti-phosphatidylcholine (aPC) IgG with arterial thrombosis. Nevertheless, the connection could not be confirmed for venous thrombosis [24]. The results of Zigon et al. established a significant association of aCL IgG, aβ2GPI IgG, and aPS/PT IgG with thrombosis. IgG, IgM, and IgA isotypes of aPS/PT, IgG, and IgA isotypes of aCL and aβ2GPI were linked to obstetric morbidity [25]. Although the triple-positive aPL profile has been described as a significant risk factor for thrombosis [26], recent findings have shown that the sole presence of LAC is superior to double- and triple-positive aPL profiles in APS secondary to SLE [27]. The value of adding non-criteria aPLs in the evaluation of thrombosis risk still requires clarification. Therefore, searching for these less-described aPLs and seeking to understand their function could make a difference.

When analysing the thrombosis risk prediction, our results indicated both criteria and non-criteria aPLs titres as predictors of different thrombotic manifestations. Some of the specific thrombotic events—especially arterial thrombosis and pregnancy-related morbidity—seemed to be rather associated with criteria aPLs, particularly with aβ2GPI IgM. Meanwhile, venous thrombotic events were apparently linked to the positivity of some non-criteria aPLs—aPT IgM, aPS IgG, or aPS IgM. Therefore, the possible use of non-criteria aPLs in thrombotic risk stratification should not be neglected.

The predictive role of some aPLs for non-thrombotic APS manifestations has been described in the literature [28]. Our data suggested aPE IgG as the best predictor for non-criteria clinical events in APS.

The aim of our research was to determine the correlation degrees of non-criteria aPLs with the criteria ones in patients diagnosed with primary and secondary APS. Significant positive correlations of the diagnostic and non-diagnostic tested antibodies were identified, suggesting a possible synchronous overall production of the aPLs.

Even though the hypothesis that the non-criteria aPL profile could represent a distinguishing mark between the primary and secondary APS patients was raised, further studies are necessary to confirm this possible role of non-classical aPLs [29]. Our results did not show any significant differences between the titres of non-criteria aPLs in primary and secondary APS.

PS is a negatively charged molecule. Antibodies against PS can be found in 70% of patients having a seropositive APS diagnosis. They can also be detected in the serum of seronegative APS patients [30]. Our results indicated aPS IgG as the most important predictor for deep venous thrombosis among the searched antibodies. Moreover, when evaluating the risk of deep venous thrombosis, associated with arterial thrombosis, and obstetric morbidity in our study population, aPS IgM proved to be a good predictor, right after aβ2GPI IgG and aβ2GPI IgM. The present study mainly revealed moderate correlations when evaluating the entire study population, the strongest being the correlations of aPS IgM with both IgM and IgG isotypes of aCL. In the primary and secondary APS subgroups, significant correlations were found between aPS IgM and aCL IgM. aCLs are independent predictors of thrombosis in APS patients [31]. Therefore, the possible link of aPS with thrombotic event occurrence should be investigated. However, since the titres of these antibodies correlated very well with the titres of criteria aPLs, especially with aCL IgM, it might not be useful in daily practice to determine these markers together.

PE belongs to the neutral phospholipid group, comprising the negatively charged phosphate group and positively charged ethanolamine. As aPLs mainly target negatively charged phospholipids, the antibodies developed against PE are less frequent [32]. Moreover, aPE positivity seems to be associated with APS clinical events, even in the absence of other serum aPLs [33,34]. The aPE IgM isotypes were more frequently found in patients with unexplained thrombosis or foetal loss [33]. Nonetheless, data sustaining a certain aPE involvement in the APS events are insufficient [32]. Our research revealed moderate aPE correlations with criteria and non-criteria aPLs in the primary APS subgroup. High values of the correlation coefficient were registered between the IgM isotype of aPE and the IgM isotype of aCL in secondary APS. Nevertheless, no significant correlation was found between aPE IgG and aCL IgG in the secondary APS subgroup. Similar to the case of aPS, the relatively good degree of correlation of aPE with criteria aPLs should raise doubts on the relevance of their combined testing.

Prothrombin (PT) is a glycoprotein, a precursor of thrombin in the coagulation cascade. Notably, aPT might compete with clotting factors during coagulation processes or might catalyse prothrombin proteolysis [35]. Recent results indicate the role of aPT in the occurrence of thrombosis in APS [23]. When evaluating the association of different antibodies with deep venous thrombosis combined with obstetric morbidity, aPT IgM was found as the best predictor. When considering the entire study group, both IgM and IgG isotypes of aPT weakly correlated with aCL IgG. Among the primary APS patients, aPT IgM or IgG isotypes were not significantly correlated with criteria aPLs (aCL IgG and aβ2GPI IgG) or non-criteria aPLs (aPS IgG). In the secondary APS subgroup, aPT IgM or IgG isotypes registered non-significant correlations with non-criteria (aPS IgG and aPE IgG) and criteria aPLs (aCL IgG, aβ2GPI IgG, and aβ2GPI IgM). One necessary remark is that the patients included in this research no longer presented thrombotic events. This might be an explanation for the relatively low degree of correlations detected between these antibodies and criteria aPLs. Furthermore, it is important to mention that aβ2GPI and aPT can have LAC activity in the presence of the coagulation factor V [36]. This could also be a reason for the weak correlation found between aPT and the rest of the tested aPLs, as the detection of LAC was not performed at the time of enrolment, due to technical reasons.

In addition to aPT, the determination of the aPS/PT complex has been cited as the most sensitive of non-criteria aPLs in patients with seropositive APS [16]. Even if aPT and aPS/PT share many similarities and might co-exist in the same patient, they belong to different antibody populations. The correlation between the aPS/PT complex and the clinical traits of APS is sustained by increasing evidence [37]. Multiple research protocols have highlighted the association of aPS/PT with LAC [38,39]. The possibility of using IgG and IgM aPS/PT as a replacement for LAC has even been suggested. This could be especially useful in patients taking oral anticoagulants, in whom the detection of LAC is challenging [38].

The aPS/PT complex was not investigated in the present study. The results obtained by separately evaluating aPT and aPS were contradictory. Although the correlation degrees obtained for aPS suggest a rather limited diagnostic utility, aPT registered the lowest values, highlighting them as the most viable path for further research.

There are certain limitations in our study. Firstly, a relatively small number of subjects were included, and only the cross-sectional evaluation of aPL profiles was performed. Moreover, the study population did not include any pure obstetric APS patients. Other important restraints were the impossibility of detecting LAC, especially considering that aβ2GPI and aPT might exert LAC activity [36], and the lack of aPL tests standardisation on an international level. The fact that none of the patients were having a thrombotic event at the time of enrolment could also represent a disadvantage. Ultimately, despite the wide-ranging detected aPL titres, most of the results were negative, with only a few found above the laboratory positivity cut-off.

As far as we know, the present research is one of very few in the literature concerning the correlations between criteria and non-criteria antibody titres in APS. Notwithstanding the reduced size of the study sample, a high number of both criteria and non-criteria aPLs were investigated. Good correlations were found overall, as well as in the subgroups, between criteria and the non-criteria aPLs, suggesting good sensitivity and specificity of the antibodies currently used in the classification criteria. Weaker degrees of correlation were detected for aPT, making them a possible future candidate for APS classification criteria.

In summary, the present research revealed significant correlations between criteria and non-criteria aPLs, suggesting that the potential inclusion of non-classical aPLs in the APS classification criteria is still a matter of debate. Considering its reduced degrees of correlation, aPT seems to be the most promising candidate for becoming a classification criterion. Future research will most probably clarify the role of non-classical aPLs, either as new APS classification criteria or as a diagnostic tool in seronegative APS.

## Figures and Tables

**Table 1 biomedicines-11-02192-t001:** Age at enrolment, age at diagnosis, and the duration of the disease in the study group and subgroups.

	Overall	Primary APS	Secondary APS	*p*-Values
Mean age at enrolment	44.94 ± 11.60 years	42.73 ± 11.00 years	46.52 ± 11.88 years	0.168
Mean age at diagnosis	38.78 ± 11.74 years	39.9 ± 10.97 years	37.98 ± 12.34 years	0.489
Duration of the disease	3 (1–29) years	2.87 ± 2.11 years	8.55 ± 7.36 years	<0.001

Data are presented as mean ± standard deviation or as median (minimum–maximum), according to their parametric or non-parametric features.

**Table 2 biomedicines-11-02192-t002:** Clinical and serological characteristics of the study subgroups.

		Primary APS	Secondary APS	*p*-Values	OR (95% CI)
Number of patients		30	42		
Gender	Male	6	4	0.21	0.42 (0.10–1.64)
	Female	24	38		
Deep venous thrombosis		14	16	0.46	0.70 (0.27–1.81)
Recurrent deep venous thrombosis		7	4	0.11	0.34 (0.09–1.31)
Arterial thrombosis		7	17	0.13	2.23 (0.78–6.36)
Recurrent arterial thrombosis		1	8	0.07	6.82 (0.80–57.82)
Obstetric morbidity		6	11	0.54	1.41 (0.45–4.38)
Deep venous thrombosis + Arterial thrombosis		0	6	0.22	3.91 (0.43–35.41)
Deep venous thrombosis + Obstetric morbidity		1	3	0.49	2.23 (0.22–22.55)
Arterial thrombosis + Obstetric morbidity		0	3	0.49	2.23 (0.22–22.55)
Deep venous thrombosis + Arterial thrombosis + Obstetric morbidity		0	2	0.56	1.05 (0.98–1.12)
Non-thrombotic manifestations (such as livedo, vegetation, migraine, convulsions)		11	14	0.76	0.86 (0.32–2.30)
Serology at the time of diagnosis	aCL	6	9	0.88	1.09 (0.34–3.47)
	aβ2GPI	9	10	0.55	0.72 (0.25–2.09)
	LAC	30	42		
	aCL + aβ2GPI	2	5	0.46	1.89 (0.34–10.47)
	aCL + LAC	6	9	0.88	1.09 (0.34–3.47)
	aβ2GPI + LAC	9	10	0.55	0.72 (0.25–2.09)

APS—antiphospholipid syndrome; aCL—anti-cardiolipin antibodies; aβ2GPI—anti-β2 glycoprotein I antibodies; LAC—lupus anticoagulant.

**Table 3 biomedicines-11-02192-t003:** Serological characteristics of the study subgroups at the time of enrolment.

aPL Profile	Primary APS—Number of Positive Patients	Secondary APS—Number of Positive Patients
aCL	1	4
aβ2GPI	4	8
aPE	4	8
aPS	1	5
aPT	2	2
aCL + aβ2GPI + aPE	0	3
aCL + aβ2GPI + aPS	0	3
aCL + aβ2GPI + aPT	0	1
aCL + aPE + aPS	1	3
aCL + aPE + aPT	0	1
aCL + aPS + aPT	0	1
aβ2GPI + aPE + aPS	0	3
aβ2GPI + aPE + aPT	0	1
aβ2GPI + aPS + aPT	0	1
aPE + aPS + aPT	0	1
aCL + aβ2GPI + aPE + aPS	0	3
aCL + aβ2GPI + aPE + aPT	0	1
aCL + aβ2GPI + aPS + aPT	0	1
aβ2GPI + aPE + aPS + aPT	0	1
aCL + aPE + aPS + aPT	0	1
aCL + aβ2GPI + aPE + aPS + aPT	0	1

APS—antiphospholipid syndrome; aPLs—antiphospholipid antibodies; aCL—anti-cardiolipin antibodies; aβ2GPI—anti-β2 glycoprotein I antibodies; aPE—anti-phosphatidylethanolamine antibodies; aPS—anti-phosphatidylserine antibodies; aPT—anti-prothrombin antibodies.

**Table 4 biomedicines-11-02192-t004:** Titres of the tested aPLs, overall and in the study subgroups.

	Overall Median Values (Min–Max)	Primary APS Median Values (Min–Max)	Secondary APS Median Values (Min–Max)	*p*-Values
aCL IgG	1 (0–121) U/mL	0 (0–121) U/mL	1 (0–64) U/mL	0.03
aCL IgM	4 (0–41) U/mL	4 (0–12) U/mL	4 (0–41) U/mL	0.45
aβ2GPI IgG	3 (0–79) U/mL	3 (0–30) U/mL	3 (0–79) U/mL	0.92
aβ2GPI IgM	6 (0–300) U/mL	6 (0–36) U/mL	6 (0–300) U/mL	0.67
aPE IgG	2 (0–151) U/mL	2 (1–20) U/mL	2 (0–151) U/mL	0.94
aPE IgM	7 (0–202) U/mL	9 (0–23) U/mL	7 (0–88) U/mL	0.93
aPS IgG	2 (0–130) U/mL	2 (0–30) U/mL	2 (1–112) U/mL	0.13
aPS IgM	4 (0–31) U/mL	4 (1–14) U/mL	4 (0–29) U/mL	0.38
aPT IgG	4 (1–33) U/mL	4 (1–33) U/mL	4 (1–20) U/mL	0.81
aPT IgM	3 (0–82) U/mL	2 (0–82) U/mL	4 (0–25) U/mL	0.59

APS—antiphospholipid syndrome; aPLs—antiphospholipid antibodies; aCL—anti-cardiolipin antibodies; aβ2GPI—anti-β2 glycoprotein I antibodies; aPE—anti-phosphatidylethanolamine antibodies; aPS—anti-phosphatidylserine antibodies; aPT—anti-prothrombin antibodies; Ig—immunoglobulin.

**Table 5 biomedicines-11-02192-t005:** The aPLs’ correlations in APS patients, overall.

	aCL IgG	aCL IgM	aβ2GPI IgG	aβ2GPI IgM	aPE IgG	aPE IgM	aPS IgG	aPS IgM	aPT IgG	aPT IgM
aCL IgG	1	0.35 **	0.51 **	0.42 **	0.40 **	0.44 **	0.72 **	0.50 **	0.26 *	0.25 *
aCL IgM	0.35 **	1	0.27 *	0.77 **	0.37 **	0.67 **	0.36 **	0.77 **	0.44 **	0.58 **
aβ2GPI IgG	0.51 **	0.27 *	1	0.37 **	0.40 **	0.44 **	0.37 **	0.50 **	0.26 *	0.34 **
aβ2GPI IgM	0.42 **	0.77 **	0.37 **	1	0.54 **	0.64 **	0.35 **	0.64 **	0.58 **	0.56 **
aPE IgG	0.40 **	0.37 **	0.40 **	0.54 **	1	0.65 **	0.56 **	0.46 **	0.61 **	0.28 *
aPE IgM	0.44 **	0.67 **	0.44 **	0.64 **	0.65 **	1	0.48 **	0.74 **	0.46 **	0.50 **
aPS IgG	0.72 **	0.36 **	0.37 **	0.35 **	0.56 **	0.48 **	1	0.52 **	0.31 **	0.21
aPS IgM	0.50 **	0.77 **	0.50 **	0.64 **	0.46 **	0.74 **	0.52 **	1	0.36 **	0.55 **
aPT IgG	0.26 *	0.44 **	0.26 *	0.58 **	0.61 **	0.46 **	0.31 **	0.36 **	1	0.36 **
aPT IgM	0.25 *	0.58 **	0.34 **	0.56 **	0.28 *	0.50 **	0.21	0.55 **	0.36 **	1

APS—antiphospholipid syndrome; aPLs—antiphospholipid antibodies; aCL—anti-cardiolipin antibodies; aβ2GPI—anti-β2 glycoprotein I antibodies; aPE—anti-phosphatidylethanolamine antibodies; aPS—anti-phosphatidylserine antibodies; aPT—anti-prothrombin antibodies; Ig—immunoglobulin; * *p* < 0.05; ** *p* < 0.01.

**Table 6 biomedicines-11-02192-t006:** The aPLs’ correlations in primary APS.

	aCL IgG	aCL IgM	aβ2GPI IgG	aβ2GPI IgM	aPE IgG	aPE IgM	aPS IgG	aPS IgM	aPT IgG	aPT IgM
aCL IgG	1	0.26	0.64 **	0.21	0.62 **	0.54 **	0.76 **	0.51 **	0.25	0.14
aCL IgM	0.26	1	0.31	0.69 **	0.46 **	0.54 **	0.46 **	0.81 **	0.38 *	0.53 **
aβ2GPI IgG	0.64 **	0.31	1	0.26	0.54 **	0.50 **	0.51 **	0.41 *	0.27	0.31
aβ2GPI IgM	0.21	0.69 **	0.26	1	0.56 **	0.56 **	0.39 *	0.61 **	0.55 **	0.55 **
aPE IgG	0.62 **	0.46 **	0.54 **	0.56 **	1	0.77 **	0.71 **	0.57 **	0.63 **	0.40 *
aPE IgM	0.54 **	0.54 **	0.50 **	0.56 **	0.77 **	1	0.56 **	0.67 **	0.50 **	0.51 **
aPS IgG	0.76 **	0.46 **	0.51 **	0.39 *	0.71 **	0.56 **	1	0.67 **	0.40 *	0.26
aPS IgM	0.51 **	0.81 **	0.41 *	0.61 **	0.57 **	0.67 **	0.67 **	1	0.39 *	0.54 **
aPT IgG	0.25	0.38 *	0.27	0.55 **	0.63 **	0.50 **	0.40 *	0.39 *	1	0.48 **
aPT IgM	0.14	0.53 **	0.31	0.55 **	0.40 *	0.51 **	0.26	0.54 **	0.48 **	1

APS—antiphospholipid syndrome; aPLs—antiphospholipid antibodies; aCL—anti-cardiolipin antibodies; aβ2GPI—anti-β2 glycoprotein I antibodies; aPE—anti-phosphatidylethanolamine antibodies; aPS—anti-phosphatidylserine antibodies; aPT—anti-prothrombin antibodies; Ig—immunoglobulin; * *p* < 0.05; ** *p* < 0.01.

**Table 7 biomedicines-11-02192-t007:** The aPLs’ correlations in secondary APS.

	aCL IgG	aCL IgM	aβ2GPI IgG	aβ2GPI IgM	aPE IgG	aPE IgM	aPS IgG	aPS IgM	aPT IgG	aPT IgM
aCL IgG	1	0.37 *	0.46 **	0.52 **	0.29	0.43 **	0.68 **	0.46 **	0.34 *	0.27
aCL IgM	0.37 *	1	0.26	0.78 **	0.35 *	0.74 **	0.31 *	0.75 **	0.48 **	0.61 **
aβ2GPI IgG	0.46 **	0.26	1	0.46 **	0.30 *	0.38 *	0.29	0.56 **	0.25	0.29
aβ2GPI IgM	0.52 **	0.78 **	0.46 **	1	0.54 **	0.67 **	0.35 *	0.68 **	0.61 **	0.54 **
aPE IgG	0.29	0.35 *	0.30 *	0.54 **	1	0.53 **	0.47 **	0.42 **	0.58 **	0.18
aPE IgM	0.43 **	0.74 **	0.38 *	0.67 **	0.53 **	1	0.43 **	0.79 **	0.38 *	0.46 **
aPS IgG	0.68 **	0.31 *	0.29	0.35 *	0.47 **	0.43 **	1	0.42 **	0.26	0.14
aPS IgM	0.46 **	0.75 **	0.56 **	0.68 **	0.42 **	0.79 **	0.42 **	1	0.34 *	0.52 **
aPT IgG	0.34 *	0.48 **	0.25	0.61 **	0.58 **	0.38 *	0.26	0.34 *	1	0.28
aPT IgM	0.27	0.61 **	0.29	0.54 **	0.18	0.46 **	0.14	0.52 **	0.28	1

APS—antiphospholipid syndrome; aPLs—antiphospholipid antibodies; aCL—anti-cardiolipin antibodies; aβ2GPI—anti-β2 glycoprotein I antibodies; aPE—anti-phosphatidylethanolamine antibodies; aPS—anti-phosphatidylserine antibodies; aPT—anti-prothrombin antibodies; Ig—immunoglobulin; * *p* < 0.05; ** *p* < 0.01.

## Data Availability

Not applicable.

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
