# Peer review of "Criteria and Non-Criteria Antiphospholipid Antibodies in Antiphospholipid Syndrome: How Strong Are They Correlated?"

_biomedicines, 2023, doi:10.3390/biomedicines11082192_

Round 1

Reviewer 1 Report

The manuscript “Criteria and non-criteria antiphospholipid antibodies in antiphospholipid syndrome – how strong are they correlated?” by Simona Caraiola et co-authors from Romania. evaluated the presence of non-criteria antiphospholipid antibodies of antiphospholipid syndrome (APS.

I think this work is interesting. The fact that non-criteria antiphospholipid antibodies are related to arterial thrombosis in patients with APS is already well known from other cohorts.

However, there are some limitations that I would like to be discussed:

•    APS criteria are well defined. But there is no information about non-criterion clinical manifestations. For example, livedo? Are patients with low titers of aPL included? Positivity for antiphospholipid antibodies is separated into low, medium, and high positive. It is unclear which cut-off is used to determine correlations with APS manifestations.

•    The authors should clearly define their inclusion and exclusion criteria: acute and chronic infections, cancer, hypertension, marked renal impairment may all impact on their measured serum and functional markers.

•    Were the patients recruited from clinics only (less severe but stable disease) or also from the wards (acute flares)? Did the rheumatologist examine the patients? What specifically are the manifestations of systemic lupus in patients? What is the activity of the disease (SLEDAI or?)?

I suggest that the authors cite: Stojanovich L, Kontic M, Smiljanic D, et all. Association between non-thrombotic neurological and cardiac manifestations in patients with antiphospholipid syndrome.  Clin Exp Rheumatol. 2013; 31 (5): 756-760.

The manuscript “Criteria and non-criteria antiphospholipid antibodies in antiphospholipid syndrome – how strong are they correlated?” by Simona Caraiola et co-authors from Romania. evaluated the presence of non-criteria antiphospholipid antibodies of antiphospholipid syndrome (APS).

The study deals with a topic that is important both for clinical practice and has a scientific contribution. This is an observational, cross-sectional, single center study, reporting a significant number of patients with APS.

There are some limitations that I would like to be discussed:

•    APS criteria are well defined. But there is no information about non-criterion clinical manifestations. For example, livedo? Are patients with low titers of aPL included? Positivity for antiphospholipid antibodies is separated into low, medium, and high positive. It is unclear which cut-off is used to determine correlations with APS manifestations.

•    The authors should clearly define their inclusion and exclusion criteria: acute and chronic infections, cancer, hypertension, marked renal impairment may all impact on their measured serum and functional markers.

•    Were the patients recruited from clinics only (less severe but stable disease) or also from the wards (acute flares)? Did the rheumatologist examine the patients? What specifically are the manifestations of systemic lupus in patients? What is the activity of the disease (SLEDAI or?)?

I thus suggest that the work be accepted after a minor revision.

Author Response

Response to Reviewer 1 Comments

Point 1: APS criteria are well defined. But there is no information about non-criterion clinical manifestations. For example, livedo? Are patients with low titers of aPL included? Positivity for antiphospholipid antibodies is separated into low, medium, and high positive. It is unclear which cut-off is used to determine correlations with APS manifestations.

Response 1: We thank the Reviewer for this feedback. For every patient, a thorough evaluation of clinical manifestations was performed at the time of enrolment and data were registered for further analysis. However, the aim of the present article was to determine the degrees of correlation between several criteria and non-criteria antiphospholipid antibodies. The association of the tested antibodies with the main clinical manifestations of antiphospholipid syndrome was evaluated using ROC analysis.

Regarding the titres of antiphospholipid antibodies, as mentioned in the manuscript, at the moment of enrolment most of the values registered within the normal range maybe because all the patients were out of thrombotic events at the time of inclusion. There were some patients with high titres, but their number was small. The cut-off for the positivity of antibodies was established by the manufacturer (Aesku Diagnostics, Wendelsheim, Germany).

Our patients had various non-criteria clinical features, such as, livedo, migraine, convulsions, vegetations. There were no statistically significant differences between the two study subgroups regarding these manifestations.

Point 2: The authors should clearly define their inclusion and exclusion criteria: acute and chronic infections, cancer, hypertension, marked renal impairment may all impact on their measured serum and functional markers.

Response 2: Only patients diagnosed with APS according to the 2006 Sydney classification criteria and admitted to the Internal Medicine Department of Colentina Clinical Hospital were included. The following exclusion criteria were applied: age under 18 years, concomitant acute infectious disease, sarcoidosis, pregnancy, or being in the first 6 months postpartum. All the enrolled patients were evaluated for traditional cardiovascular risk factors (hypertension, diabetes mellitus, obesity, smoking). The values of creatinine and creatinine clearance were also analysed. The results indicated that, regardless of the stages of chronic kidney disease, no statistically significant differences could be found between the titres of the tested antibodies. All the data can be provided, if necessary.

Point 3: Were the patients recruited from clinics only (less severe but stable disease) or also from the wards (acute flares)? Did the rheumatologist examine the patients? What specifically are the manifestations of systemic lupus in patients? What is the activity of the disease (SLEDAI or?)?

Response 3: The patients did not present any acute flares at the time of the assessment. All the patients were evaluated by the rheumatologist at the time of enrolment. In the secondary antiphospholipid syndrome subgroup, 28 patients were having a systemic lupus erythematosus diagnosis. In 20 of these patients the SLEDAI scores were mild (0-5). A number of 6 patients registered moderate scores (0-6), while only 2 patients had severe disease activity (SLEDAI scores 13-20). The median value of SLEDAI scores was 2 (0-22). Details regarding systemic lupus erythematosus disease activity were added in the manuscript (lines 164-167).

Point 4: I suggest that the authors cite: Stojanovich L, Kontic M, Smiljanic D, et all. Association between non-thrombotic neurological and cardiac manifestations in patients with antiphospholipid syndrome.  Clin Exp Rheumatol. 2013; 31 (5): 756-760.

Response 4: The suggested source has now been cited (lines 397-399, 592-593).

Reviewer 2 Report

Authors evaluated the correlation between criteria and non-criteria aPLs in order to the potential utility of the non- classical aPLs for the diagnosis of APS.

 Although this manuscript is potentially interesting, several issues arise.

 Authors should emphasize the risk of thrombosis.

 Abstract is not clear. Take home message should be added.

 Are many APL tests standardized?

 ROC analysis may be helpful.

 Did authors examine lupus anticoagulant?

 It may helpful that authors make a algorism for the diagnosis of APS using many aPLs.

 Many aPLs should be classified in accordance with clinical use.

 Authors should discuss the clinical practice using many aPLs.

Author Response

Response to Reviewer 2 Comments

Point 1: Authors should emphasize the risk of thrombosis.

Response 1: We thank the Reviewer for this feedback. The association of antiphospholipid antibodies with thrombotic events has been performed by ROC analysis, and the results have been included in the manuscript (lines 394-396, 412-416, 444-4446, 507-510). Some of the specific thrombotic events – especially arterial thrombosis and pregnancy related morbidity – seem rather to be associated with criteria antiphospholipid antibodies, particularly with anti-β2 glycoprotein I IgM. Meanwhile, venous thrombotic events are apparently linked to the positivity of some non-criteria antiphospholipid antibodies – anti-prothrombin IgM, anti-phosphatidylserine IgG, or anti-phosphatidylserine IgM.

Point 2: Abstract is not clear. Take home message should be added.

Response 2: The abstract has been revised, according to your suggestions.

Point 3: Are many APL tests standardized?

Response 3: For this study, the ELISA kits produced by Aesku Diagnostics, Wendelsheim, Germany were used. Efforts are being made internationally for the standardization of antiphospholipid antibodies tests. However, the topic is far from being clarified and still requires significant future research. Lack of tests standardization has now been mentioned among the limitations of our paper (lines 486-487).

Point 4: ROC analysis may be helpful.

Response 4: ROC analysis evaluating the utility of criteria and non-criteria antiphospholipid antibodies in predicting the antiphospholipid syndrome subtype – primary or secondary – has been performed (Figure 1, Table 5). The results indicate IgG isotype of antiprothrombin antibodies as the best predictor for antiphospholipid syndrome subtype – 0.517 Area Under the Curve (0.377-0.658).

Point 5: Did authors examine lupus anticoagulant?

Response 5: Unfortunately, as mentioned in the manuscript, due to several logistical limitations, lupus anticoagulant could not be detected at the time of enrolment.

Point 6: It may helpful that authors make a algorism for the diagnosis of APS using many aPLs.

Response 6: The aim of this article was to detect the degrees of correlation between the criteria and several non-criteria antiphospholipid antibodies. Therefore, our paper did not intend to create a diagnostic algorithm for the antiphospholipid syndrome using more antibodies, but rather to evaluate the potential utility of the non-classical antiphospholipid antibodies for the diagnosis of the antiphospholipid syndrome. We thank you for your suggestion. It is an interesting idea. As far as we know, until the moment there has not been described a diagnostic algorithm for antiphospholipid syndrome.

Point 7: Many aPLs should be classified in accordance with clinical use.

Response 7: The association of non-criteria antiphospholipid antibodies with various clinical manifestations of the antiphospholipid syndrome has been lately emphasised by an impressive number of studies. At the time of enrolment in the study, data regarding clinical manifestations of the disease have been carefully collected for every patient. ROC analysis of the risk of different clinical events in relation to the titres of the tested antibodies was performed. When considering the deep venous thrombosis, the most important predictor seems to be aPS IgG. For arterial thrombosis the highest area under the curve was registered for aβ2GPI IgM, closely followed by aPT IgG. The best predictor of obstetric pathological events among the analysed antibodies proved to be aβ2GPI IgM. The risk of associating deep venous thrombosis with arterial thrombosis was best predicted of aβ2GPI IgM, the risk of developing deep venous thrombosis associated with obstetric morbidity seemed to be best predicted by aPT IgM, while for the risk of arterial thrombosis associated with obstetric morbidity aβ2GPI IgM registered the highest area under the curve. When evaluating the risk of deep venous thrombosis associated with arterial thrombosis and obstetric morbidity, aβ2GPI IgG, aβ2GPI IgM, and aPS IgM proved to be the best predictors. However, since the purpose of this article did not consist of determining the association between antiphospholipid antibodies and the clinical manifestations of the syndrome, the various specific clinical events that led to the diagnosis were not detailed in this paper.

Point 8: Authors should discuss the clinical practice using many aPLs.

Response 8: As mentioned before, the present article intended to evaluate whether the non-criteria antiphospholipid antibodies correlate or not with the criteria ones, deeming that the least correlated non-classical antibodies could be candidates for becoming a classification criterium for the antiphospholipid syndrome. Even so, we have evaluated the association of different thrombotic events with antiphospholipid antibodies in APS, and the results can be found in the manuscript (lines 394-396, 412-416, 444-4446, 507-510).

Round 2

Reviewer 2 Report

Authors has relatively responded the comments.

However, revised manuscript is still not clear nor attractive.

It is helpful to remake many figures and table.

Author Response

Response to Reviewer 2 Comments

We thank you very much for the time dedicated to the manuscript “Criteria and non-criteria antiphospholipid antibodies in antiphospholipid syndrome – how strong are they correlated?”.

We would like to express our gratitude for the assessment of our manuscript, as well as for the valuable suggestions made in your review notes, and for providing the opportunity of a revision.

Therefore, we have revised once again the document and made the corrections suggested by the reviewer. We have re-evaluated our responses to your previous remarks. The text of the manuscript has been extensively improved, and the tables have been revised.

Please find below a point-by-point response to your comments.

Point 1: Authors should emphasize the risk of thrombosis.

Response 1: We thank the Reviewer for the comments. We have assessed and better explained the associations of the tested antiphospholipid antibodies with thrombosis. ROC analysis has been used for evaluating the association of antiphospholipid antibodies with thrombotic events, and the results have been included in the manuscript (lines 498-507, 519-523, 548-550). Tables 7 to 13 present the exact values of areas under the curve that have been obtained. Some of the specific thrombotic events – especially arterial thrombosis and pregnancy related morbidity – seem rather to be associated with criteria antiphospholipid antibodies, particularly with anti-β2 glycoprotein I IgM. Meanwhile, venous thrombotic events are apparently linked to the positivity of some non-criteria antiphospholipid antibodies – anti-prothrombin IgM, anti-phosphatidylserine IgG, or anti-phosphatidylserine IgM.

Point 2: Abstract is not clear. Take home message should be added.

Response 2: Thank you very much for your comment. The abstract has been revised once again. We believe that the aim of the study and the study conclusions are more clearly expressed know, and we are opened for any further suggestions for improving the abstract.

Point 3: Are many APL tests standardized?

Response 3: For this study, the ELISA kits produced by Aesku Diagnostics, Wendelsheim, Germany were used. Efforts are being made internationally for the standardization of antiphospholipid antibodies tests. However, the topic is far from being clarified and still requires significant future research. Lack of tests standardization on an international level has now been mentioned among the limitations of our paper (lines 705-706).

Point 4: ROC analysis may be helpful.

Response 4: ROC analysis evaluating the utility of criteria and non-criteria antiphospholipid antibodies in predicting the antiphospholipid syndrome subtype – primary or secondary – has been performed (Figure 1, Table 6). The results indicate IgG isotype of anti-prothrombin antibodies as the best predictor for antiphospholipid syndrome subtype – 0.517 Area Under the Curve (0.377-0.658). Furthermore, the association of the tested antiphospholipid antibodies titres with different thrombotic events (Tables 7-13), and with non-thrombotic clinical manifestations of antiphospholipid syndrome (Table 14) has been assessed using the ROC analysis.

Point 5: Did authors examine lupus anticoagulant?

Response 5: Thank you for this observation. Due to several logistical limitations, lupus anticoagulant could not be detected at the time of enrolment. We have mentioned that “Technical reasons did not allow testing the LAC, since the samples could not be processed as fast as the determination protocol required.” (lines 142-144). However, it is important to mention that all the enrolled patients were already having an established antiphospholipid syndrome diagnosis when they were recruited for this study. At the time when they were diagnosed, all the patients have had their lupus anticoagulant evaluated and were all positive for lupus anticoagulant. The serological characteristics of the study population at the time of diagnosis have been presented in the manuscript, in Table 3. Our study protocol did not include re-testing for lupus anticoagulant because of the previously mentioned reasons. The serological profiles of the patients, obtained at the enrolment, comprising the anti-cardiolipin antibodies, anti-β2 glycoprotein I antibodies, anti-phosphatidylethanolamine antibodies, anti-phosphatidylserine antibodies, and anti-prothrombin antibodies, have been presented in Table 4, while the titres are listed in Table 5.

Although the determination of the lupus anticoagulant is a very important test for the antiphospholipid syndrome diagnosis, it was not of interest for our study, as lupus anticoagulant is not an antiphospholipid antibody per se. Many antiphospholipid antibodies (best known for domain I of the anti-β2 glycoprotein I) have actually intrinsec lupus anticoagulant activity, and so we might have had a bias when assessing correlations with lupus anticoagulant.

Point 6: It may helpful that authors make a algorism for the diagnosis of APS using many aPLs.

Response 6: Thank you for this interesting output. The aim of this article was to detect the degrees of correlation between the criteria and several non-criteria antiphospholipid antibodies. Therefore, our paper did not intend to create a diagnostic algorithm for the antiphospholipid syndrome using more antibodies, but rather to evaluate the potential utility of the non-classical antiphospholipid antibodies for the diagnosis of the antiphospholipid syndrome. We thank you for your suggestion. We will consider it for further reserch in which non-APS cases could be also included.

Point 7: Many aPLs should be classified in accordance with clinical use.

Response 7: We thank you for this observation. We have tried to clarify all these aspects. The association of non-criteria antiphospholipid antibodies with various clinical manifestations of the antiphospholipid syndrome has been lately emphasised by an impressive number of studies. At the time of enrolment in the study, data regarding clinical manifestations of the disease have been carefully collected for every patient. ROC analysis of the risk of different clinical events in relation to the titres of the tested antibodies was performed (Tables 7-14). When considering the deep venous thrombosis, the most important predictor seems to be anti-phosphatidylserine IgG. For arterial thrombosis the highest area under the curve was registered for anti-β2 glycoprotein I IgM, closely followed by anti-prothrombin IgG. The best predictor of obstetric pathological events among the analysed antibodies proved to be anti-β2 glycoprotein I IgM. The risk of associating deep venous thrombosis with arterial thrombosis was best predicted of anti-β2 glycoprotein I IgM, the risk of developing deep venous thrombosis associated with obstetric morbidity seemed to be best predicted by anti-prothrombin IgM, while for the risk of arterial thrombosis associated with obstetric morbidity anti-β2 glycoprotein I IgM registered the highest area under the curve. When evaluating the risk of deep venous thrombosis associated with arterial thrombosis and obstetric morbidity, anti-β2 glycoprotein I IgG, anti-β2 glycoprotein I IgM, and anti-phosphatidylserine IgM proved to be the best predictors. However, since the purpose of this article did not consist of determining the association between antiphospholipid antibodies and the clinical manifestations of the syndrome, the various specific clinical events that led to the diagnosis were not detailed in this paper. In regard to the possible role of non-criteria antiphospholipid antibodies in the diagnosis of antiphospholipid syndrome, our results indicated anti-prothrombin antibodies as the most promising candidate for becoming a new classification criterion, since they were the least correlated with the criteria antiphospholipid antibodies.

Point 8: Authors should discuss the clinical practice using many aPLs.

Response 8: Thank you very much for your comment. As mentioned before, the present article intended to evaluate whether the non-criteria antiphospholipid antibodies correlate or not with the criteria ones, deeming that the least correlated non-classical antibodies could be candidates for becoming classification criteria for the antiphospholipid syndrome. Our results identified mainly moderate correlations between the tested non-criteria antibodies and the criteria ones, suggesting little added value for the use of the tested non-criteria antiphospholipid antibodies, with the exception of anti-prothrombin antibodies, that seem to have different kinetics and might represent a future classification criterion for the antiphospholipid syndrome. In addition, we have evaluated the association of different thrombotic events with antiphospholipid antibodies in APS, and the results can be found in the manuscript (lines 498-507, 519-523, 548-550).

Round 3

Reviewer 2 Report

Thank you for your full responses to my comments.

However, there are many tables. It is better that several tables move to supplementary tables.

Several tables can be combined.

Author Response

Response to Reviewer 2 Comments

We are grateful for your precious time invested in reviewing the paper “Criteria and non-criteria antiphospholipid antibodies in antiphospholipid syndrome – how strong are they correlated?”.

We highly appreciate your insightful suggestions, and we are sincerely thankful for the opportunity of a revision.

We have carefully considered your comments. The manuscript has been revised once again, and we have done our utmost to address every correction suggested by the reviewer.

We present below our point-by-point response to your comments.

Point 1: Thank you for your full responses to my comments. However, there are many tables. It is better that several tables move to supplementary tables.

Response 1: Thank you very much for your comment. All the tables presenting the results of ROC analysis in regard to the utility of antiphospholipid antibodies in predicting the antiphospholipid syndrome subtype, different specific thrombotic events, or non-thrombotic antiphospholipid syndrome events (Tables 6-14) have been relocated to the Supplementary Material section – Supplementary Tables S1 – S5.

Point 2: Several tables can be combined.

Response 2: We thank you for this observation. Tables 2 and 3 have been combined in one table presenting the clinical and serological characteristics of the study population. Tables 7, 10, and 11 have been combined into one table presenting the utility of criteria and non-criteria aPLs in predicting deep venous thrombosis, deep venous thrombosis associated with arterial thrombosis, and deep venous thrombosis associated with obstetric morbidity – Supplementary Table S2. Furthermore, Tables 8, 12, and 13 have been compiled into a single table presenting the utility of criteria and non-criteria aPLs in predicting arterial thrombosis, arterial thrombosis associated with obstetric morbidity, and arterial thrombosis associated with deep venous thrombosis and obstetric morbidity – Supplementary Table S3.